# Concurrent gliomas in patients with multiple sclerosis

Katharina Sahm [1,2✉], Tobias Kessler[3,4], Philipp Eisele[1], Miriam Ratliff [5], Elena Sperk[6], Laila König[7], Michael O. Breckwoldt [2,8], Corinna Seliger[4,9], Iris Mildenberger[1,2], Daniel Schrimpf[10,11], Christel Herold-Mende[12], Pia S. Zeiner[13], Ghazaleh Tabatabai[14], Sven G. Meuth[15], David Capper [16,17], Martin Bendszus[8], Andreas von Deimling [10,11], Wolfgang Wick [3,4], Felix Sahm [10,11] & Michael Platten [1,2✉]

## Abstract

**Background** Concurrent malignant brain tumors in patients with multiple sclerosis (MS) constitute a rare but paradigmatic phenomenon for studying neuroimmunological mechanisms from both molecular and clinical perspectives.

**Methods** A multicenter cohort of 26 patients diagnosed with both primary brain tumors and multiple sclerosis was studied for disease localization, tumor treatment-related MS activity, and molecular characteristics specific for diffuse glioma in MS patients.

**Results** MS neither predisposes nor protects from the development of gliomas. Patients with glioblastoma WHO grade 4 without isocitratdehydrogenase (IDH) mutations have a longstanding history of MS, whereas patients diagnosed with IDH-mutant astrocytoma WHO grade 2 receive multiple sclerosis diagnosis mostly at the same time or later. Concurrent MS is associated with a lesser extent of tumor resection and a worse prognosis in IDH-mutant glioma patients (PFS 32 vs. 64 months, $p = 0.0206$). When assessing tumor-intrinsic differences no distinct subgroup-defining methylation pattern is identified in gliomas of MS patients compared to other glioma samples. However, differential methylation of immune-related genetic loci including human leukocyte antigen locus on 6p21 and interleukin locus on 5q31 is found in MS patients vs. matched non-MS patients. In line, inflammatory disease activity increases in 42% of multiple sclerosis patients after brain tumor radiotherapy suggesting a susceptibility of multiple sclerosis brain tissue to pro-inflammatory stimuli such as ionizing radiation.

**Conclusions** Concurrent low-grade gliomas should be considered in multiple sclerosis patients with slowly progressive, expansive T2/FLAIR lesions. Our findings of typically reduced extent of resection in MS patients and increased MS activity after radiation may inform future treatment decisions.

### Plain Language Summary

Brain tumors such as gliomas can evade attacks by the immune system. In contrast, some diseases of the central nervous system such as multiple sclerosis (MS) are caused by an overactive immune system. Our study looks at a cohort of rare patients with both malignant glioma and concurrent MS and examines how each disease and their treatments affect each other. Our data suggest that even in patients with known MS, if medical imaging findings are unusual, a concurrent brain tumor should be excluded at an early stage. Radiotherapy, as is the standard of care for malignant brain tumors, may worsen the inflammatory disease activity in MS patients, which may be associated with certain genetic risk factors. Our findings may help to inform treatment of patients with brain tumors and MS.

A full list of author affiliations appears at the end of the paper.

Gliomas have been reported in patients with multiple sclerosis (MS)[1,2]. As immune surveillance is critical in cancer development, a possible causal immuno-genetic relationship between neuroinflammatory diseases and malignant brain tumors has been repeatedly discussed[3]. Proposed cancer-promoting mechanisms include a malignant transformation of multiple sclerosis plaques[4] and immunosuppression induced by immunomodulatory therapies[5–7]. However, two population-based studies could not verify an altered risk for gliomas in patients with MS[8,9].

From a clinical perspective, concurrent brain tumors may interfere with the diagnosis and course of MS and *vice versa*. MS lesions can mimic cerebral malignancies in conventional magnetic resonance imaging (MRI)[10], and misdiagnosing brain tumor symptoms as relapse in patients with known MS can delay appropriate tumor treatment[2]. Moreover, due to the limited number of cases and lack of detailed analyses of clinical and imaging disease patterns, outcome, and molecular profiling, little is known about the effects of radio- and chemotherapy on MS disease activity and progression.

Discovering the role of epigenetic regulations such as histone modification or DNA methylation for susceptibility, progression, and treatment response of brain tumors has resulted in a fundamental shift in molecular classification and subtyping[11,12]. In contrast to distinct brain tumor methylomes, epigenetic changes in non-neoplastic diseases are subtler. In MS, aberrant DNA methylation patterns affect disease development and pathogenesis by mediating inflammatory response and neurodegeneration[13]. Interestingly, methylation differences are not only seen in pathologically altered tissue but can also be recapitulated in normal-appearing white matter or peripheral blood cells of MS patients and mainly occur in immune-related genes[14].

While the failure of immune surveillance is crucial for pathogenesis in both conditions, the complex interplay between the immune microenvironment, genetic alterations, and epigenetic regulation is only partly understood.

In this study, we analyzed clinical, imaging, and molecular data of a multicenter cohort of 26 patients with MS and concurrent glioma to determine if brain tumors in MS patients are molecularly distinct and if concurrent MS affects tumor prognosis and occurrence of treatment-associated adverse events.

We show that patients with MS are not at increased risk for malignant gliomas, but the prognosis of patients with IDH-mutant gliomas and concurrent MS is worse compared to patients without concurrent inflammatory CNS disease. Moreover, brain tumor radiotherapy can trigger inflammatory disease activity in MS patients, which may be associated with a distinct DNA methylation pattern of immune-related genetic loci.

## Methods

**Study population.** In this multicenter, retrospective analysis, we collected 30 brain tumor tissue samples diagnosed between 01 January 2001 and 31 December 2017 and clinical and radiographical data of patients with reported multiple sclerosis from the Departments of Neuropathology, Neurology and Neurosurgery, University of Heidelberg; the Departments of Neurology and Neurosurgery, University Medical Center Mannheim of Heidelberg University; Dr Senckenberg Institute of Neurooncology, University of Frankfurt, and the Department of Neurology, University of Tübingen (all located in Germany). Four cases were excluded due to tumor histology other than glioma or insufficient proof of multiple sclerosis diagnosis according to the revised McDonald criteria[15] resulting in a study population of 26 patients in the experimental cohort. Control cases with isocitrate dehydrogenase (IDH)-mutant 1p19q-intact astrocytoma WHO grade 2 or IDH-wildtype glioblastoma WHO grade 4 were collected

from the Departments of Neuropathology, Neurology, and Neurosurgery, University of Heidelberg, and matched for tumor histology and the known prognostic markers IDH- and O6-methylguanine-DNA methyltransferase (MGMT)-status, age at diagnosis, and Karnofsky-Index. Tissue sample collection and data collection and use were performed after patients provided written informed consent following local ethics regulations and approved by the local ethics committee (ethics committee of Heidelberg University, Medical Faculty Heidelberg and ethics committee of Heidelberg University, Medical Faculty Mannheim, ethics votes 005/2003, 2013-832R-MA, 2018-614N-MA).

**Clinical assessments.** Progression-free survival (PFS), defined as the time from brain tumor surgery to the date of tumor progression, was assessed based on follow-up imaging performed under the response assessment in neuro-oncology (RANO) criteria[16,17]. The overall survival (OS) was defined as the time from surgery to the date of decease. Multiple sclerosis disease activity was assessed based on clinical attacks and MRI follow-up according to the revised McDonald criteria[15].

**Methylation analysis.** Bioinformatic analysis of Illumina 450k methylation array data was performed with customized scripts adapted from the R (www.r-project.org) based library ChAMP (version: 2.10.116) as described previously[18]. At the Genomics and Proteomics Core Facility of the German Cancer Research Center (DKFZ) in Heidelberg, Germany, the Infinium HumanMethylation450 BeadChip and kits (Illumina, San Diego, CA, USA) were used according to the manufacturer's instructions to obtain the DNA methylation status at >450,000 sites in paraffin-embedded tissue derived from glioma patients with MS and matched controls. Raw data was loaded using the load() function in ChAMP. For multi-hit sites, SNPs, and XY chromosome-related CpGs data was filtered; next, data was normalized with a BMIQ-based method and analyzed for batch effects with a singular value decomposition (SVD) algorithm. There was no significant batch to be corrected. Respective functions were used to assess differentially methylated regions (DMR). Principle component analysis was used for dimensionality reduction. Circos plots showing DMR distribution were constructed with the package OmicCircos (version: 1.26.0).

**Statistical analyses.** Logistic regression analyses to calculate odds ratios (ORs) with 95% confidence intervals (CIs) were performed to compare clinical characteristics in the experimental cohort and control cohort. Distribution of survival times was estimated by the Kaplan–Meier method and compared between groups with the log-rank test. A *p*-value of less than 0.05 was considered significant.

**Reporting summary.** Further information on research design is available in the Nature Portfolio Reporting Summary linked to this article.

## Results

**Clinical characteristics of patients with multiple sclerosis and concurrent glioma.** A multicenter, retrospective cohort of 26 patients with both MS and diffuse glioma, meeting the predefined inclusion criteria was assembled from four centers (Fig. 1, Supplementary Data File 1, Supplementary Data 1). A survey of 2809 brain tumor patients (C71 ICD-10-GM), diagnosed between 01 January 2007 and 31 December 2015, revealed no evidence of increased MS prevalence compared to published cohorts with or without autoimmune diseases (Anssar et al.[8]) (Supplementary Table 2). All gliomas had an astrocytic morphology, with IDH-mutant astrocytomas WHO grade 2 being the most frequent brain tumors in MS patients (46%), followed by IDH-wildtype

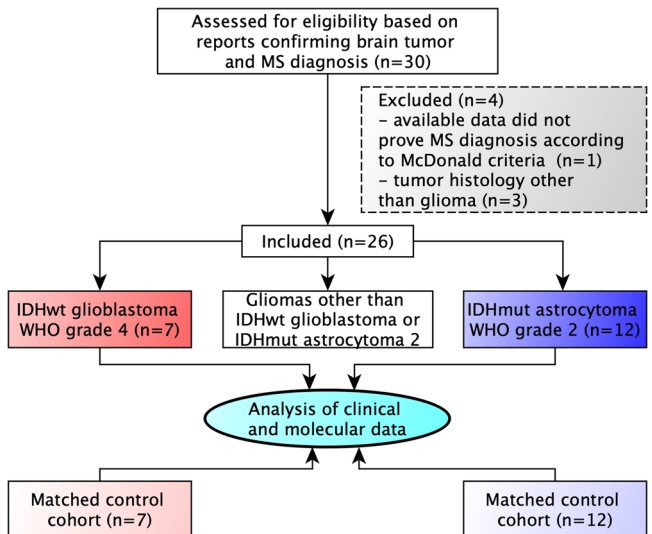

**Fig. 1 Flow diagram of the multicenter study.** Diagram showing the number and main inclusion and exclusion criteria of recruited patients as well as group allocation.

glioblastomas WHO grade 4 (31%). Median age at tumor diagnosis was 32.5 years for IDH-mutant astrocytomas and 59 years for glioblastomas, both within the age range expected for these tumor types[19] (Fig. 2a, Supplementary Tables 3, 4). The mean age at MS diagnosis was 31 years and 40 years, respectively (Fig. 2b, Supplementary Table 5). The MS symptom course was relapsing-remitting in 19/26 (73%) cases and secondary progressive in 4/26 (15%). All patients had radiologically proven cerebral inflammatory lesions, 6/26 (23%) had spinal lesions, and 7/26 (27%) had optic neuritis. 15/26 (58%) patients received immunomodulatory treatment (Supplementary Data 1, Supplementary Fig. 1B).

While patients diagnosed with glioblastoma had a longstanding history of CNS inflammatory disease, patients diagnosed with IDH-mutant astrocytoma received MS diagnosis mostly at the same time or later. Subsequently, glioblastomas arose near existing inflammatory lesions, while IDH-mutant astrocytomas developed from radiologically normal-appearing white matter (NAWM) (Fig. 2d–f, Supplementary Fig. 2).

**Concurrent multiple sclerosis diagnosis is associated with worse prognosis of IDH-mutant astrocytomas.** Next, clinical data from glioma patients with concurrent MS were compared

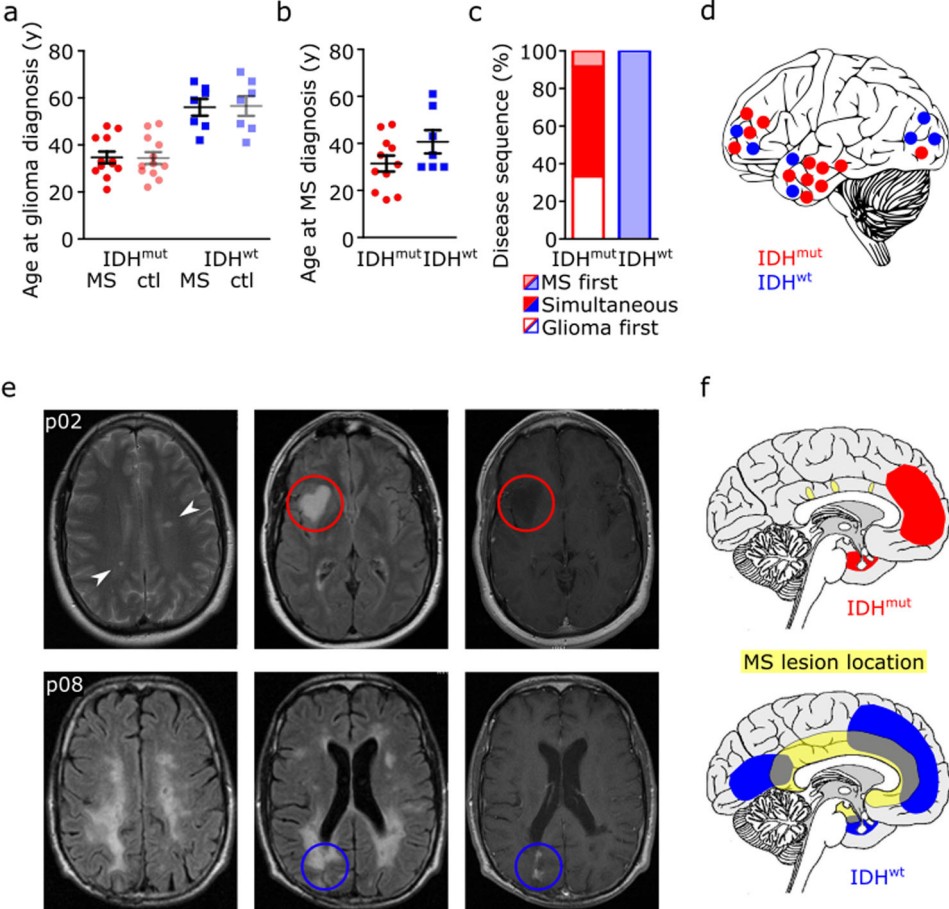

**Fig. 2 Clinical and radiological characteristics of 19 patients with multiple sclerosis and concurrent glioma. a, b** Scatter dot plot of patients' age at initial histologically proven diagnosis of glioma depending on the mutation status of isocitrate dehydrogenase-1 (IDH) in patients with concurrent multiple sclerosis (MS) or control patients without multiple sclerosis (ctl) (**a**), and patients' age at initial multiple sclerosis diagnosis depending on tumoral IDH mutation status (**b**). Depicted are single and mean values with SEM (IDH-mut MS n = 12; IDH-wt MS n = 7). **c** Chronology of multiple sclerosis and glioma diagnosis depending on tumor histology judged by IDH mutation status. **d** Schematic representation of glioma location in patients with concurrent multiple sclerosis depending on tumoral IDH mutation status. Dots represent single patients. **e** MRI with axial FLAIR and contrast-enhanced T1 images of a patient with multiple sclerosis and concurrent IDH-mutant astrocytoma WHO grade 2 (p02, upper row) and of a patient with multiple sclerosis and IDH-wildtype glioblastoma WHO grade 4 (p08, lower row). **f** Schematic representation of the spatial relation of multiple sclerosis lesions and IDH-mutant astrocytomas (red) or IDH-wildtype glioblastomas (blue) in patients with concurrent diseases.

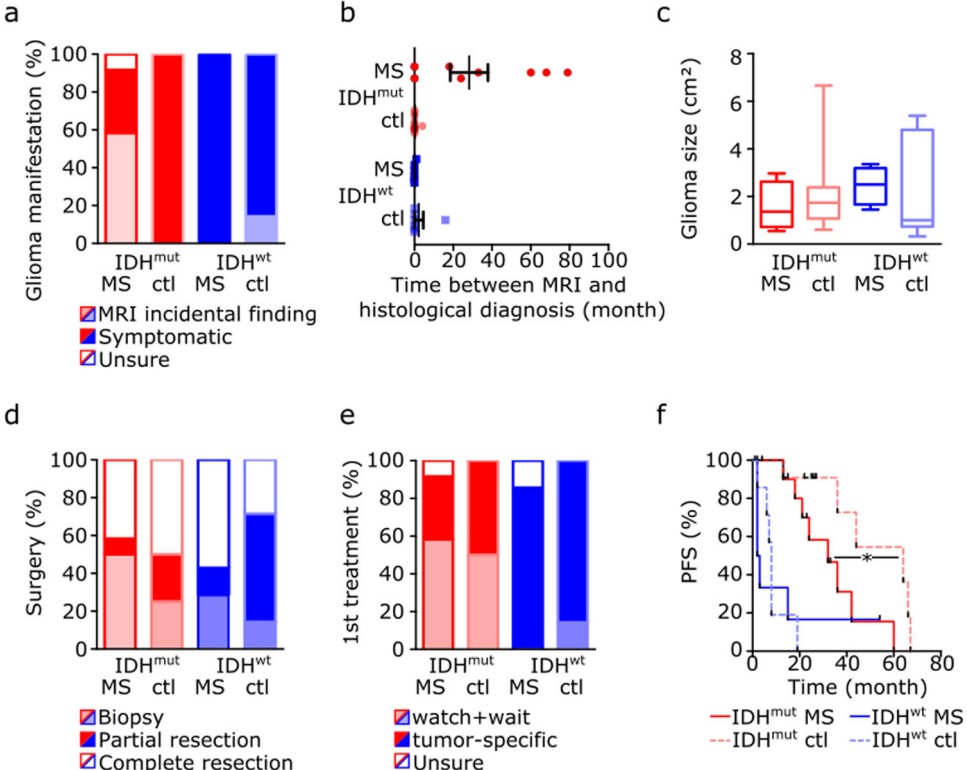

**Fig. 3 Clinical course of brain tumor disease in patients with glioma and concurrent multiple sclerosis. a** Percentage of patients with or without glioma-caused neurological symptoms leading to initial tumor diagnosis in patients with concurrent multiple sclerosis (MS) or matched controls (ctl) depending on tumoral isocitrate dehydrogenase-1 (IDH) mutation status. **b** Time between radiological tumor suspicion and subsequent histological diagnosis. Depicted are single and mean values with SEM (IDH-mut MS $n = 10$; IDH-mut ctl $n = 12$; IDH-wt MS $n = 6$; IDH-wt ctl $n = 7$). **c** Tumor size as detected by MRI at the time of histological diagnosis in patients with concurrent multiple sclerosis or matched controls depending on tumoral IDH mutation status. Displayed are the products of the two largest perpendicular diameters ($cm^2$) in axial FLAIR or contrast-enhanced T1 images for IDH-mutant astrocytomas WHO grade 2 and IDH-wildtype glioblastomas, respectively. Depicted are mean values with SEM and min to max (IDH-mut MS $n = 8$; IDH-mut ctl $n = 8$; IDH-wt MS $n = 2$; IDH-wt ctl $n = 7$). **d** Percentage of patients receiving tumor biopsy, partial, or complete resection. **e** Type of adjuvant tumor therapy depending on concurrent multiple sclerosis diagnosis and IDH mutation status. **f** Kaplan–Meier analysis of progression-free survival (PFS) after primary tumor treatment depending on IDH status and concurrent multiple sclerosis diagnosis (IDH-mut MS $n = 12$; IDH-mut ctl $n = 12$; IDH-wt MS $n = 7$; IDH-wt ctl $n = 7$).

with data from control patients without MS and matched for tumor histology and the known prognostic markers IDH- and MGMT-status, age at diagnosis, and Karnofsky-Index (Supplementary Tables 3, 4). The glioblastoma-related neurological symptoms and distinct radiological morphology led to prompt radiological tumor suspicion and subsequent histological diagnosis independent of MS comorbidity. In contrast, radiological findings in most MS patients with IDH-mutant astrocytoma WHO grade 2 occurred incidentally, with a median time of 21 months (0–68) between initial MRI suspicion and histological diagnosis (Fig. 3a, b). However, glioma size at the time of histological diagnosis was comparable between MS and control groups for both tumor types (Fig. 3c). For non-contrast enhancing lesions, results from non-invasive MRI spectroscopy could support tumor suspicion in one patient (Supplementary Fig. 3). Glioma diagnosis was confirmed by biopsy in 14/26 (54%) cases, and 12/26 (46%) received partial or complete tumor resection. Remarkably, the proportion of patients receiving only a biopsy and no tumor resection tended to be higher in MS patients than controls independent of tumor histology (29% vs. 14% for IDH-wildtype; 50% vs. 25% for IDH-mutant; $p = 0.23$) (Fig. 3d). While all glioblastoma patients received standard tumor-specific treatment, except one non-MS patient receiving best supportive care, the majority of patients with IDH-mutant astrocytomas were primarily treated with a watchful waiting approach according to the current standard of care independent of MS diagnosis

(Fig. 3e). As expected[20], patients with IDH-wildtype glioblastomas showed a significantly shorter PFS compared to patients with IDH-mutant astrocytomas independent of concurrent MS. Of note, multiple sclerosis was associated with a worse PFS in IDH-mutant tumors only (PFS 32 vs. 64 months, $p = 0.0206$) (Fig. 3f) with a consistent trend for OS, although reliable statistical evaluation was not possible due to limited availability of OS data in the control group (Supplementary Fig. 1D). The limited sample size did also not allow to firmly establish the impact of immunomodulatory treatments on brain tumor progression (Supplementary Fig. 1E).

**Glioma radiotherapy as a possible risk factor for multiple sclerosis disease activity.** The majority of patients were treated with radiotherapy (RT) alone or in combination with chemotherapy at one time during the course of brain tumor disease (83% in the IDH-mutant and 71% in the IDH-wildtype cohort) (Fig. 4a, Supplementary Fig. 1C). While none of the IDH-mutant astrocytoma patients showed clinical or radiological signs of multiple sclerosis disease activity according to NEDA-3 criteria within the 12 months prior to starting RT, 5/12 patients (42%) with multiple sclerosis and IDH-mutant astrocytoma showed disease activity within 12 months after RT, with an average of 1.4 clinical relapses. Remarkably, 89% of clinical episodes occurred within the first six months after RT and were associated with new

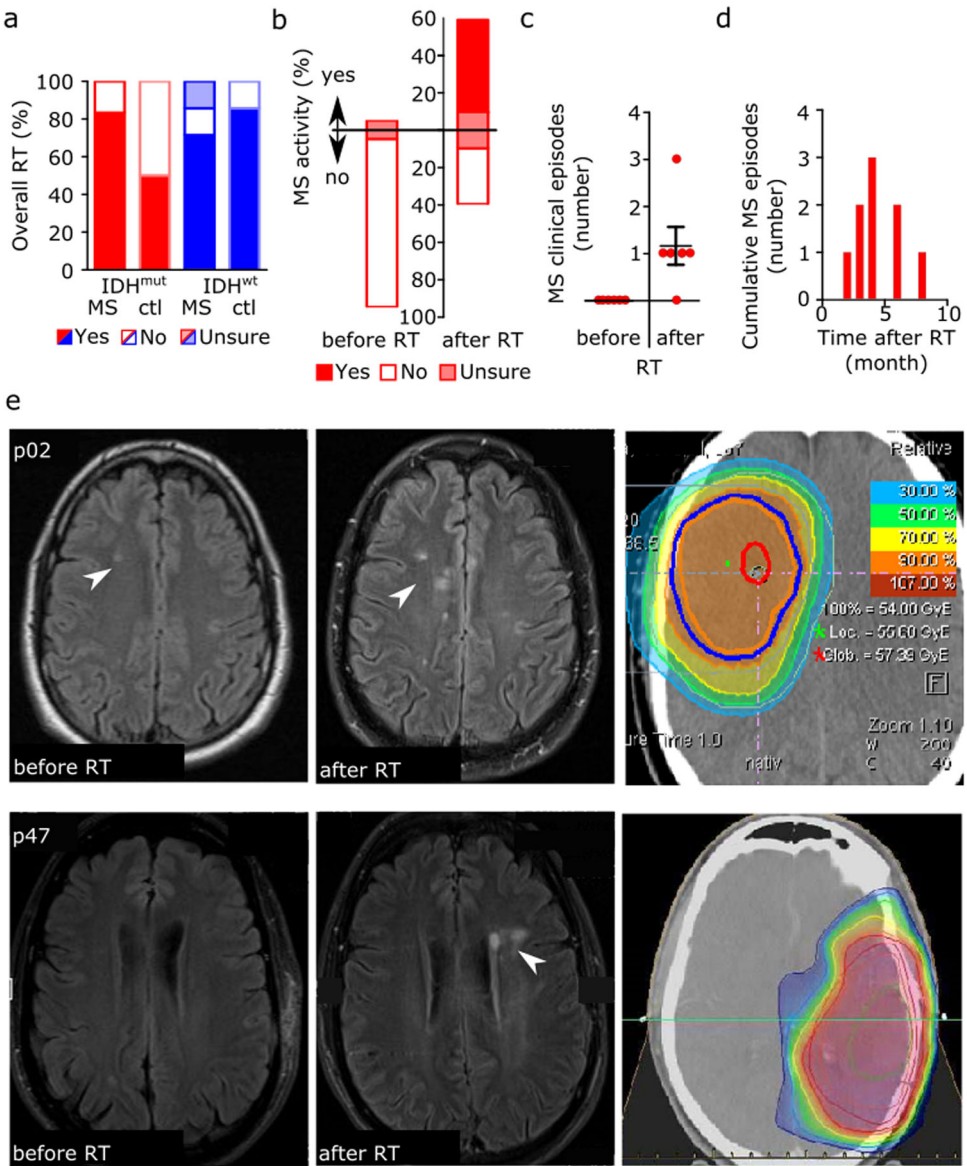

**Fig. 4 Association between radiotherapy and multiple sclerosis disease activity. a** Percentage of patients with or without radiotherapy (RT) treatment during the course of brain tumor disease depending on concurrent multiple sclerosis (MS) diagnosis and tumoral IDH mutation status. **b** Percentage of patients with or without clinical or radiological MS disease activity in the 12 months preceding (left bar) and after RT (right bar). **c** Scatter dot plot of the number of clinical episodes per patient for six multiple sclerosis patients in the 12 months preceding and after RT. Depicted are single and mean values with SEM. **d** Time course of cumulative clinical multiple sclerosis episodes for six multiple sclerosis patients experiencing disease activity in the 12 months after RT. Individual number of clinical episodes per patient are one episode in 4/6 patients, two episodes in 1/6 patients and three episodes in 1/6 patients. **e** Magnet resonance imaging (MRI) with axial FLAIR images of two patients with isocitrate dehydrogenase-1 (IDH)-mutant astrocytomas showing new demyelinating lesions (arrows) after RT within the 50% isodose line (p02 photon irradiation 60/2 Gy; p47 proton irradiation 54/1.8 Gy).

contrast-enhancing lesions within or adjacent to the radiation field without being associated to a certain isodose level (Fig. 4, Supplementary Fig. 1F). In 3/5 patients, radiation-associated neuroinflammation occurred despite immunomodulatory treatment, and in 2/5 patients, multiple sclerosis was initially diagnosed within six months after RT. Interestingly, one patient with IDH-wildtype glioblastoma experienced long-term survival with no evidence of tumor recurrence for eight years following radiation-induced pseudo-progression without any further antitumor therapy. The radiographical increase in disease activity was accompanied by neurological deterioration, leading to the initial diagnosis of a chronic progressive multiple sclerosis course eight months after completion of radiotherapy (Fig. 5). In both control

groups, there was no imaging evidence of neuroinflammation within the 12 months after radiotherapy.

**Genome-wide methylation analysis of glioma samples identified DMRs associated with neuroinflammation.** Genome-wide methylation patterns of glioma tissue from 12 multiple sclerosis patients and a matched control cohort without multiple sclerosis were compared to address the question if chronic neuroinflammation may predispose susceptibility to the exacerbation of focal disease activity on a molecular level in multiple sclerosis patients after radiotherapy (Supplementary Tables 3, 4). While principal component analysis and unsupervised clustering revealed no distinct subgroup-defining methylation pattern depending on multiple

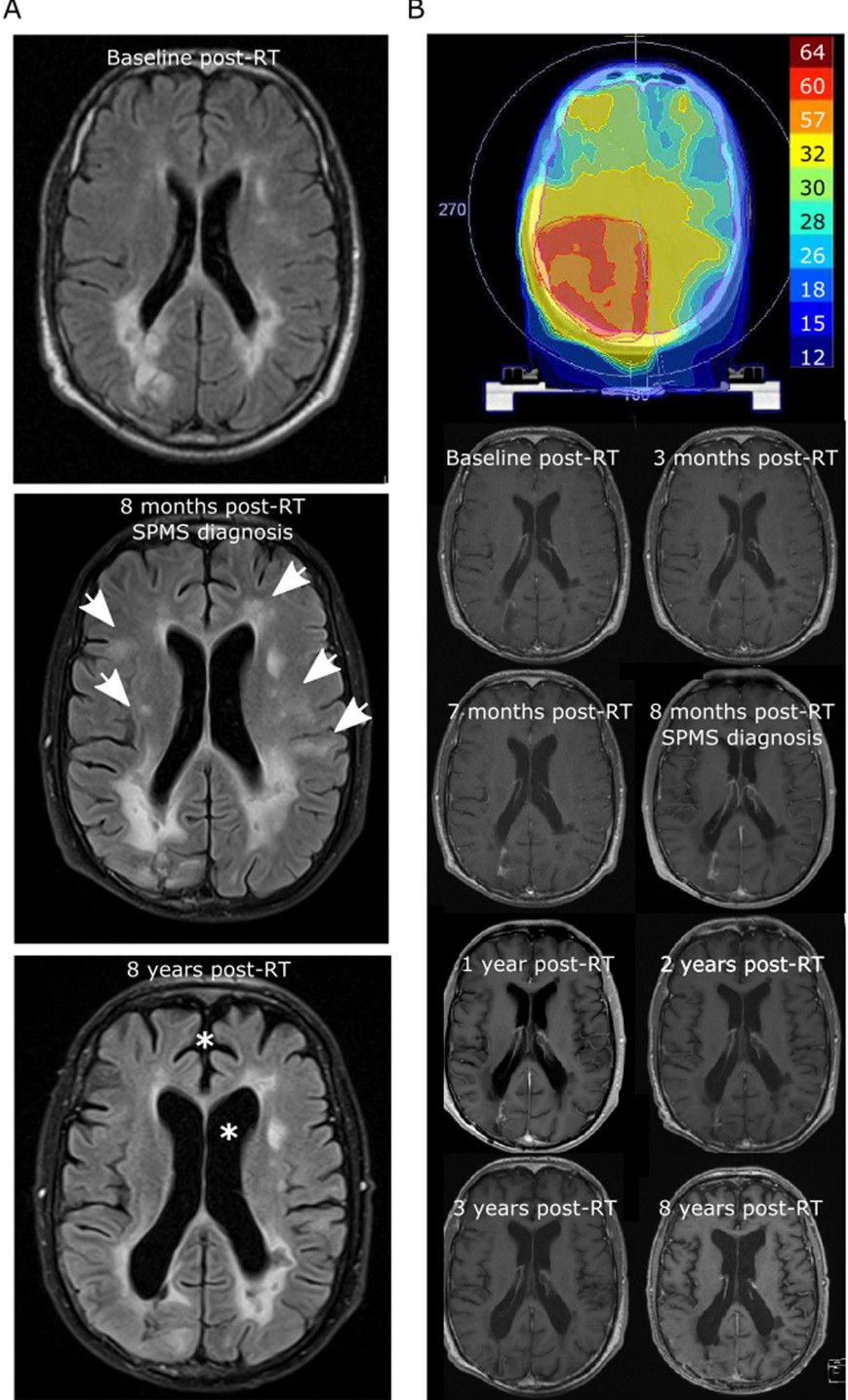

**Fig. 5 Long-term response after radiation-induced pseudoprogression.** Representation of isodoses (Gy) and longitudinal MRIs with axial FLAIR images (**A**) and contrast-enhanced axial T1 images (**B**) of a patient (p08) with isocitrate dehydrogenase-1 (IDH)-wildtype glioblastoma WHO grade 4. Transient contrast enhancement within the tumor area corresponding to radiation-induced pseudoprogression with a maximum of 8 months after completion of radiotherapy (60/2 Gy) is associated with an increase in subcortical multiple sclerosis lesions (arrows) and the initial diagnosis of a chronic progressive multiple sclerosis course. Long term follow-up shows a moderate generalized brain atrophy (stars) without evidence of glioblastoma recurrence up to 8 years after initial tumor therapy.

sclerosis diagnosis, analysis of DMRs identified 117 DMRs in IDH-mutant astrocytomas and 103 DMRs in IDH-wildtype glioblastomas, with 11 common DMRs possibly associated with multiple sclerosis (Fig. 6a, Supplementary Fig. 4). Interestingly, DMRs build a hotspot on the short arm of chromosome 6 (6p21.32, 6p22.1) within

the human leukocyte antigen (HLA) region (6p22.1-21.3) with the majority of affected regions being hypomethylated compared to control groups (Fig. 6b). The HLA system plays a critical role in immune regulation by presenting epitopes to leukocytes and thereby releasing antigen-specific immune responses[21]. For multiple

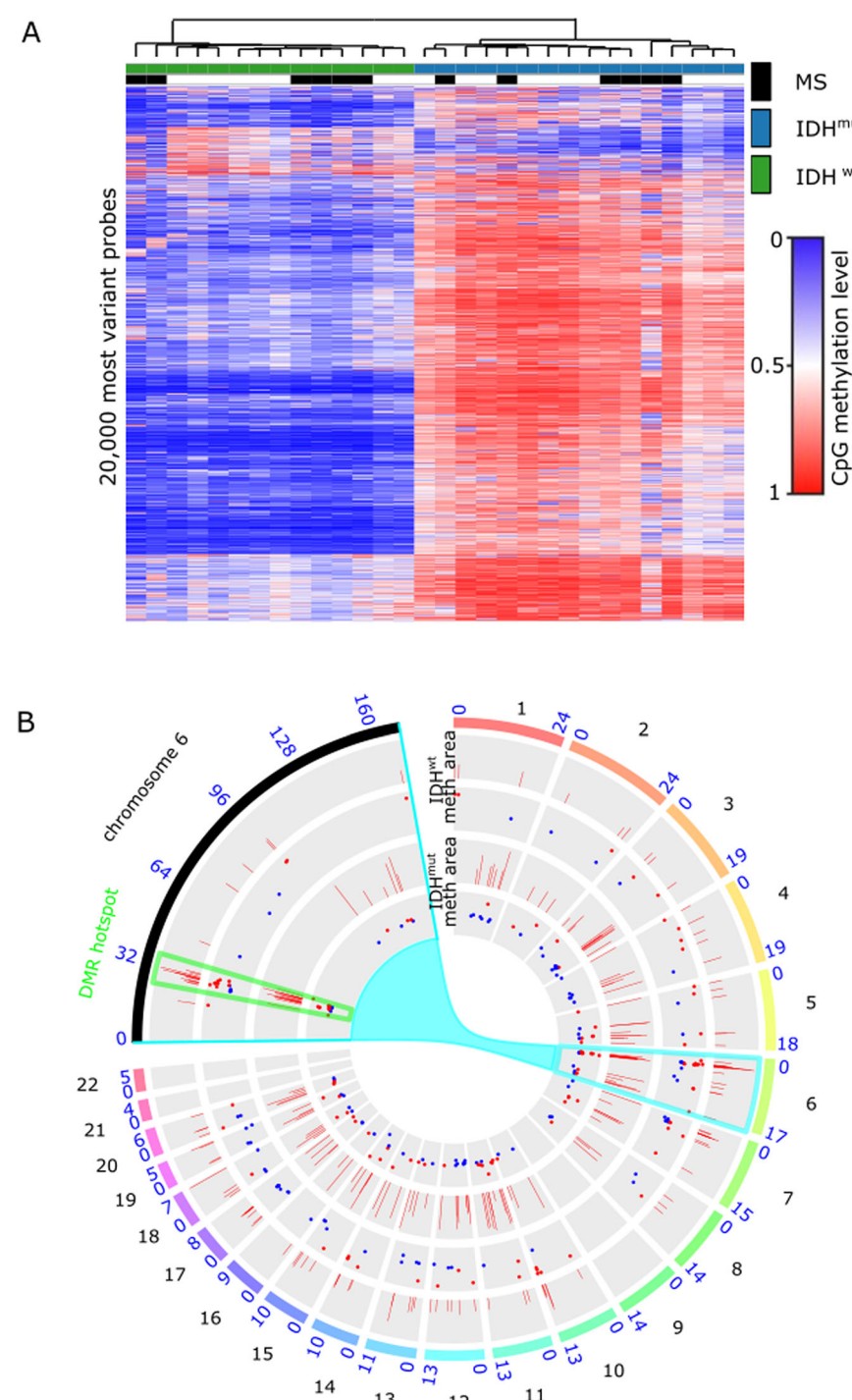

**Fig. 6 Genome-wide DNA methylation changes between gliomas of patients with concurrent multiple sclerosis and controls. A** Unsupervised hierarchical clustering of methylation profiles from 450k methylation array data of 12 glioma tissue in multiple sclerosis (MS) patients and matched controls. **B** Circos plot showing differentially methylated regions (DMR) between patients with and without multiple sclerosis separately for isocitrate dehydrogenase-1-mutant (IDH^mut, inner two circles) and isocitrate dehydrogenase-1-wildtype tumors (IDH^wt, outer two circles) on each of the human chromosomes with a zoom in on chromosome 6. The inner circle of the respective comparisons show the average methylation, the outer circle the area of the DMR.

sclerosis, the HLA system has been shown to comprise major susceptibility loci (Gourraud et al.[22]), and the aberrant methylation patterns of specific HLA genes in NAWM of multiple sclerosis patients are thought to be relevant for multiple sclerosis etiology[14]. Moreover, most known multiple sclerosis risk genes belong to the immune system, and genetic regions related to innate immunity are

thought to constitute glioma risk factors[3]. In line, additional DMRs are located on chromosome 5q31 within the interleukin (IL) region and include genes associated with myelinization and immune function (Supplementary Table 6). These data suggest that epigenetic changes in inflammatory genes in NAWM predispose to exacerbation of neuroinflammation by focal irradiation.

## Discussion

Multiple sclerosis with concurrent glioma is a rare phenomenon, with only a few cases reported[9,23–25]. From an immunological perspective, both diseases comprise opposed paradigmatic conditions within the central nervous system (CNS), with excessive immune reactions causing multiple sclerosis and profound immunosuppression enabling glioma progression, respectively[26–28]. The role of underlying onco-immunological mechanisms for disease susceptibility and etiology has been repeatedly discussed without answering the question of an incidental co-existence or causal relationship[23,29,3]. In this context, epigenetic mechanisms could play an essential role. While region-specific changes in DNA methylation occur in multiple sclerosis brains[14], the methylome is known as a diagnostic and predictive factor in primary brain tumors[3,12]. By comparative analysis of genome-wide DNA methylation patterns of gliomas occurring in patients with and without concurrent multiple sclerosis, we could identify DMRs involving immune-relevant genes such as HLA and interleukin regions (Fig. 6). Although the pathophysiological relevance remains unproven, these results support previous findings of subtle but widespread and persistent epigenetic changes in the CNS tissue of multiple sclerosis patients[14] and could reflect underlying (subclinical) neuroinflammation. However, one major limitation of the presented study is the small sample and subgroup size which could also explain why no distinct subgroup-defining methylation pattern was identified. In addition, due to rarity of this condition, validation of identified genes in the regions affected by epigenetic changes in an independent patient population was not possible so far.

One major clinical challenge is the accurate detection of brain malignancies in patients with known multiple sclerosis, where especially early non-enhancing gliomas could be misdiagnosed as multiple sclerosis lesions. On the other hand, tumefactive multiple sclerosis lesions can mimic brain tumors when viewed with conventional MRI techniques[2]. In our cohort, the vast majority of IDH-mutant astrocytomas in patients suffering from multiple sclerosis were incidentally detected during routine radiological follow-up. Although an average of almost two years passed until histological validation was realized, MRI tumor size at the time of surgery was comparable to that of the control group in which further diagnostic procedures were triggered by neurological symptoms. Remarkably, a relevant proportion of multiple sclerosis patients received tumor biopsy only without further resection, which might have contributed to an unfavorable prognosis (Fig. 3). Although the small sample sizes constitute a major limitation, our data suggest the importance of an additional diagnostic work-up, the early biopsy of atypical or suspicious lesions, and in case of a brain tumor diagnosis a second surgery to allow for (partial) resection as a known prognostic marker.

So far, little is known about the disease course of multiple sclerosis patients with concurrent brain tumors. While at the early stages of relapsing-remitting multiple sclerosis, clinical episodes associated with inflammatory lesions largely recover, incomplete recovery causes secondary disease progression over time[30]. Although some case series suggest a positive impact of brain tumors on multiple sclerosis disease course[31], conclusive data including the role of brain cancer treatments and multiple sclerosis disease-modifying therapies are lacking. Our data support the concept of radiation-induced inflammation in multiple sclerosis, with close to half the patients presenting with clinical or radiological signs of disease activity after glioma radiotherapy, while showing no evident disease activity before radiotherapy (Fig. 4). Based on our clinical and epigenetic study, patients with known MS should be monitored closely during and after brain tumor RT in order to detect inflammatory disease activity and to differentiate MS episodes from tumor (pseudo)progression. Moreover, our study may contribute to define predictive markers for radiation-induced neuroinflammation from tumor tissue and thus identify patients at risk to develop radiation-induced pseudoprogression. Interestingly, one patient experienced a complete long-term response after radiation-induced pseudo-progression (Fig. 5). The role of secondary immune effects including release of tumor antigens for the response to radiotherapy is well known[32], and a combination of radiotherapy and immunotherapy has shown synergistic effects also in brain tumors[33]. Thus, based on this paradigmatic concurrence of immunologically opposed CNS diseases, our data could support the concept of combined radioimmunotherapy to treat brain cancer.

## Data availability

Source data for Figs. 2–4 can be found in Supplementary Data File 1 and Supplementary Tables 1-5. Methylation data are deposited at the Gene Expression Omnibus data repository (accession number GSE243465).

## Code availability

Source code is described in Supplementary Data 2.

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

## Acknowledgements

This work was supported by a grant from the Deutsche Forschungsgemeinschaft (DFG) within CRC1366 "Vascular Control of Organ Function", project number 39404578, to K.S. and M.P. (project C1) and the Hertie Foundation (P1200013) to K.S. M.O.B. acknowledges funding by the Else Kröner-Fresenius Stiftung (2019_EKMS.23).

## Author contributions

Conception and design: K.S. and M.P. Acquisition of data: K.S., T.K., P.E., M.R., E.S., L.K., M.O.B., C.S., I.M., D.S., C.H.M., P.S.Z., G.T., S.G.M., D.C., M.B., A.vD., W.W., F.S., and M.P. Analysis of data: K.S., T.K., D.S., F.S., and M.P. Interpretation of data: K.S., T.K., P.E., M.R., E.S., L.K., M.O.B., C.S., I.M., D.S., C.H.M., P.S.Z., G.T., S.G.M., D.C., M.B., A.vD., W.W., F.S., and M.P. Writing, review, and/or revision of the manuscript: K.S., T.K., P.E., M.R., E.S., L.K., M.O.B., C.S., I.M., D.S., C.H.M., P.S.Z., G.T., S.G.M., D.C., M.B., A.vD., W.W., F.S., and M.P.

## Funding

## Competing interests

The authors declare no competing interests.

## Additional information

[1]Department of Neurology, Medical Faculty Mannheim, Mannheim Center for Translational Neurosciences (MCTN), University of Heidelberg, Mannheim, Germany. [2]Clinical Cooperation Unit Neuroimmunology and Brain Tumor Immunology, German Cancer Consortium (DKTK), German Cancer Research Center (DKFZ), Heidelberg, Germany. [3]Clinical Cooperation Unit Neurooncology, German Cancer Consortium (DKTK), German Cancer Research Center (DKFZ), Heidelberg, Germany. [4]Department of Neurology and Neurooncology Program, National Center for Tumor Diseases (NCT), Heidelberg University Hospital, Heidelberg, Germany. [5]Department of Neurosurgery, Medical Faculty Mannheim, Mannheim Center for Translational Neurosciences (MCTN), University of Heidelberg, Mannheim, Germany. [6]Department of Radiation Oncology, Medical Faculty Mannheim, University of Heidelberg, Mannheim, Germany. [7]Department of Radiation Oncology, Heidelberg University Hospital, Heidelberg, Germany. [8]Department of Neuroradiology, Heidelberg University Hospital, Heidelberg, Germany. [9]Wilhelm Sander-NeuroOncology Unit and Department of Neurology, Regensburg University Hospital, Regensburg, Germany. [10]Department of Neuropathology, Heidelberg University Hospital, Heidelberg, Germany. [11]Clinical Cooperation Unit Neuropathology, German Cancer Consortium (DKTK), German Cancer Research Center (DKFZ), Heidelberg, Germany. [12]Division of Experimental Neurosurgery, Department of Neurosurgery, Heidelberg University Hospital, Heidelberg, Germany. [13]Dr Senckenberg Institute of Neurooncology, University of Frankfurt, Frankfurt, Germany. [14]Department of Neurology & Interdisciplinary Neurooncology, University Hospital Tübingen, Hertie Institute for Clinical Brain Research, Eberhard Karls University Tübingen, Tübingen, Germany. [15]Department of Neurology, Institute of Translational Neurology, University Hospital Münster, Münster, Germany. [16]Department of Neuropathology, Charité - Universitätsmedizin Berlin, corporate member of Freie Universität Berlin, Humboldt-Universität zu Berlin, and Berlin Institute of Health, Berlin, Germany. [17]German Cancer Consortium (DKTK), Partner Site Berlin, German Cancer Research Center, Heidelberg, Germany. ✉email: k.sahm@dkfz.de; m.platten@dkfz.de

