## [Peer Review File · Communications Medicine]

This manuscript has been previously reviewed at another Nature Portfolio journal. This document only contains reviewer comments and rebuttal letters for versions considered at Communications Medicine

Responses to the Reviewers`comments:

Reviewer #1

The authors present findings from their large (for concurrent MS and glioma) cohort. From an MS perspective, I think the observations reinforce what is already known, specifically that this is rare and that radiotherapy can cause an inflammatory reaction. I am not sure that this would have a material effect on MS clinical practice. I think the genetic insights may be of interest to neuro-oncologists. I think the paper is clearly written, but wonder if the discussion could bring out more of the relevance of the genetic findings for glioma management?

We thank the reviewer for the valuable feedback. As suggested, we have now discussed the relevance of our clinical and epigenetic findings not only for the treatment of brain tumor patients with known MS but also for the identification of patients at higher risk of developing radiation-induced tumor pseudoprogression.

Reviewer #2

In the present study, Sahm et al. aim at describe multiple sclerosis patients presenting with concurrent gliomas, specifically IDHmut astrocytoma WHO grade II and IDHwt glioblastoma WHO IV. Despite the difficulty of collect data on such a rare cohort of patients, the authors were able to present clinical data relevant to advance our knowledge in the neuroinflammatory response after RT in MS patients and show interesting observations that might be useful to consider in the management of those patients. Indeed, the final conclusion is of potentially of high interest, suggesting a benefit for brain tumor patients in the use of combined radioimmunotherapy. Overall, this is an informative observational study and interesting approach.

We thank the reviewer for the positive evaluation of our manuscript and the valuable comments. In the following, we would like to answer each of the reviewer's comments:

Major comments:

-The idea of studying MS genetic and epigenetic alterations through gliomas characteristics, and the immunoinflammatory response in gliomas using MS features, is brilliant. Unfortunately, no distinct subgroup-defining methylation pattern was identified in gliomas of MS patients compared to other glioma samples. Indeed, in Fig 6, the authors explore the differences in the context of differential methylation regions in patient with MS + gliomas vs non-MS + gliomas, but DMRs analysis shows differences in diverse glioma histological type only, independently of the MS diagnosis. It is possible that this was an underpowered study, with small sample numbers. This limitation should be mentioned in the discussion.

-Did you validate any of the described genes in the region affected by epigenetic changes in other patients with same characteristics (Suppl Table 6; such as RNF39 and PRRT1)? Do you have any other complementary data to show for orthogonal validation?

We completely agree with the reviewer's comment that due to the small sample size our study might have been underpowered in terms of identifying subgroup-defining DNA methylation patterns. As suggested, this limitation is now discussed in greater detail. Additionally, the validation of our epigenetic data, which is not yet available due to the rarity of this condition and consequently the lack of an independent cohort, is now discussed as further limitation.

Minor comments:

-provide a table with a list of patients used in the study (p47,p08 etc) comprehensive of the molecular characterization (IDHmut/wt), MS status (+/-) and treatments for each patient.

A tabular overview of all patients has now been added (Supplementary Table 1).

-provide MRIs of more patients where possible.

As requested, we have included additional MRI images of two patients with known MS at the time of initial diagnosis of an IDH-wildtype glioblastoma (Supplementary Figure 2)

-Fig 4 and Suppl Fig 1F: clarify molecular characteristics of p47 (IDHmut?) in the legends.

We accordingly updated both legends.

-Fig 5 looks very appealing but anecdotal. What was the therapeutic strategy used for that patient? Did he receive immunotherapy after RT?

The patient did not receive any further treatment after radiotherapy leading to complete remission for eight years. The therapeutic strategy has now been clarified in the results section.

Reviewer #3

In their manuscript, Sahm et al investigate gliomas that occur in patients with multiple sclerosis (MS), including clinical characteristics, prognosis, and methylation, with a focus on immune-related features. This is a rare phenomenon, with their study focused on 26 patients with both conditions. The prevalence of MS in this group was not higher than other cohorts and the gliomas identified were astrocytic histology, either grade II or IV. IDH mutant tumors were diagnosed in younger patients and WT tumors in older patients, both as expected. GBM patients had a longstanding MS history while IDH mutant astrocytoma patients were diagnosed with MS near the time of diagnosis (or later). They show that MS patients with IDH mutant tumors had a significantly worse PFS. They also show increased MS activity following RT. Follow up methylation analysis showed no distinct subgroup defining methylation pattern dependent on MS diagnosis using PCA or unsupervised clustering, but did show some differentially methylated regions (11 common between IDH mut/wildtype GBM, including the HLA region which the authors suggest may contribute to exacerbation of neuroinflammation by focal irradiation. The findings are interesting but likely more relevant to neuro-oncology community without further validation of the molecular findings or mechanistic understanding.

We thank the reviewer for the valuable feedback. Data addressing the reviewer's specific comments have now been included:

- The authors describe differences in proportions of patients receiving interventions such as tumor biopsy and resection (3D). However, this is not done with any statistical testing to establish these are significantly different or not.

Results of statistical testing have been added accordingly.

- The authors should also show the overall survival differences for MS patients with and without IDH mutation (similar to 3F). The supplementary figure is confusing as S1D legend says PFS in the description but shows OS in the panel, whereas 1E shows PFS in the figure and in the legend says OS. I am assuming S1E was switched with S1D. Are these OS differences significant? The term "Prognosis" is vague and not specific to PFS or OS without additional description.

We thank the reviewer for this comment. Indeed, S1E was switched with S1D which has been corrected and OS is depicted in S1D. Unfortunately, reliable statistical evaluation of OS differences was not possible due to limited availability of OS data in the control group with more than 50% of OS data being censored. This explanation has now been added in the results section. Additionally, the term "Prognosis" has been specified.

- This line needs to be clarified: "Glioma diagnosis was confirmed by biopsy in 14/26 (54%) cases, and

12/26 (46%) received partial or complete tumor resection.” Reading this further in the discussion it makes it seem biopsy means no resection. This should be clarified in the results and in the figure “biopsy only”. As suggested, definition of surgical extent has been clarified in the main text.

- The authors use the term “radiation-associated neuroinflammation” in line 240, page 8. What is the frequency of these episodes in the control population? How clear is the distinction between radiological signs of MS vs. radiation associated neuroinflammation solely due to radiation?

Evaluation of MRI scans and definition of (new) MS lesions were done according to the MAGNIMS guidelines (Rovira, À. et al. Nat. Rev. Neurol. 11, 471–482 (2015)). We agree with the reviewer that these criteria do not take into account coincident structural brain lesions such as malignant gliomas, however, using this definition neither radiological signs of neuroinflammation nor comparable clinical events occurred in any patient of the control group which is now mentioned in the main text.

- If these episodes are being attributed to the radiation treatment, what was the frequency in patients who received chemo + RT or RT alone? Also were the radiographic signs of MS activity always within a certain isodose level?

We agree with the reviewer’s comment that the impact of chemotherapy for treatment-associated MS activity should also be considered. However, while the majority of patients with IDHmut astrocytoma were initially treated with a watch and wait strategy, in our GBM cohort, in each case only a single patient was treated with radiotherapy alone or chemotherapy alone, while 5/7 patients received combined radiochemotherapy according to the current guidelines. Therefore, no reliable statement can be made from our data about the role of chemotherapy for treatment-associated MS activity which is now discussed in more detail. The lesions described were located within the irradiation field but could not be attributed to a specific isodose level which is now also described in more detail in the text.

- Are figures 4C and 4D normalized per patient? Or in particular for 4D is it showing some cases where a single patient had multiple episodes.

2/5 patients experienced multiple MS episodes within 12 months after brain tumor radiotherapy. We now included this information on the number of MS episodes per single patient in the legend of Figure 4D.

- More detail should be provided for the chromosome 6 DMR near HLA. How close is it to HLA genes? is it present in the promoter? Is it increased or decreased methylation compared to controls?

We accordingly added details on the exact chromosomal location and methylation characteristics in the main text.

REVIEWERS' COMMENTS:

Reviewer #3 (Remarks to the Author):

The authors have appropriately addressed reviewer comments and added important clarifications to their manuscript including in terms of limitations and statistical testing.